# Natural Compounds: Co-Delivery Strategies with Chemotherapeutic Agents or Nucleic Acids Using Lipid-Based Nanocarriers

**DOI:** 10.3390/pharmaceutics15041317

**Published:** 2023-04-21

**Authors:** Patrícia V. Teixeira, Eduarda Fernandes, Telma B. Soares, Filomena Adega, Carla M. Lopes, Marlene Lúcio

**Affiliations:** 1CF-UM-UP—Centro de Física das Universidades do Minho e Porto, Departamento de Física da Universidade do Minho, 4710-057 Braga, Portugal; patriciavilel@hotmail.com (P.V.T.); eduardabfer@gmail.com (E.F.); telmabsoares@gmail.com (T.B.S.); 2CytoGenomics Lab, Department of Genetics and Biotechnology, University of Trás-os-Montes e Alto Douro, 5000-801 Vila Real, Portugal; filadega@utad.pt; 3BioISI-Biosystems and Integrative Sciences Institute, Faculty of Sciences, University of Lisbon, 1749-016 Lisbon, Portugal; 4FFP-I3ID—Instituto de Investigação, Inovação e Desenvolvimento, FP-BHS—Biomedical and Health Sciences Research Unit, Faculdade de Ciências da Saúde, Universidade Fernando Pessoa, Rua Carlos da Maia 296, 4200-150 Porto, Portugal; 5Associate Laboratory i4HB—Institute for Health and Bioeconomy, Faculty of Pharmacy, University of Porto, 4050-313 Porto, Portugal; 6UCIBIO—Applied Molecular Biosciences Unit, MEDTECH—Medicines and Healthcare Products, Laboratory of Pharmaceutical Technology, Department of Drug Sciences, Faculty of Pharmacy, University of Porto, 4050-313 Porto, Portugal; 7CBMA—Centro de Biologia Molecular e Ambiental, Departamento de Biologia, Universidade do Minho, 4710-057 Braga, Portugal

**Keywords:** cancer, conventional therapy, combined therapy, lipid-based nanocarriers, natural compounds

## Abstract

Cancer is one of the leading causes of death, and latest predictions indicate that cancer- related deaths will increase over the next few decades. Despite significant advances in conventional therapies, treatments remain far from ideal due to limitations such as lack of selectivity, non-specific distribution, and multidrug resistance. Current research is focusing on the development of several strategies to improve the efficiency of chemotherapeutic agents and, as a result, overcome the challenges associated with conventional therapies. In this regard, combined therapy with natural compounds and other therapeutic agents, such as chemotherapeutics or nucleic acids, has recently emerged as a new strategy for tackling the drawbacks of conventional therapies. Taking this strategy into consideration, the co-delivery of the above-mentioned agents in lipid-based nanocarriers provides some advantages by improving the potential of the therapeutic agents carried. In this review, we present an analysis of the synergistic anticancer outcomes resulting from the combination of natural compounds and chemotherapeutics or nucleic acids. We also emphasize the importance of these co-delivery strategies when reducing multidrug resistance and adverse toxic effects. Furthermore, the review delves into the challenges and opportunities surrounding the application of these co-delivery strategies towards tangible clinical translation for cancer treatment.

## 1. Introduction

According to the World Health Organization (WHO), cancer is a serious public health problem around the world, being the leading cause of mortality and causing more than 6 million deaths yearly [1,2]. While the cancer mortality rate has declined in recent years, WHO estimates it will reach 13.1 million cancer-related deaths by 2030 [3,4]. Despite extensive development of cytotoxic agents, current therapy approaches for cancer are still ineffective [5]. There are two major treatment options available: surgical procedures or non-surgical therapy regimens [5]. The surgical intervention is limited by the tumor’s size as well as the stage of metastasis in the tissues and organs from the site of origin. Non-surgical treatment options primarily include chemotherapy and radiotherapy, or a combination of these approaches [6,7]. Even though chemotherapeutic agents have evidenced efficacy in killing cancer cells by interfering with the process of cell division [5], they still face a number of challenges, including low bioavailability and lack of selectivity. Consequently, non-specific body distribution of chemotherapy is a key factor for cancer patient mortality, followed by chemo-resistance of cancer cells, which is another significant barrier that must be overcome in order to provide effective cancer treatment [2,3,7,8,9].

Several strategies have been employed to improve the performance of chemotherapeutic agents and, as a result, overcome the abovementioned challenges. Among these strategies are chemical modification, the development of new chemotherapeutic agents that are not detected by multidrug resistance (MDR) efflux pumps, and the combination of the cytotoxic agent with a chemosensitizer. Moreover, nanocarriers have been proposed to surpass some of the chemotherapy challenges. In this regard, nanocarriers for drug delivery are designed to reach specific organs and act selectively on the target site, providing advantages over conventional chemotherapeutics [1,7,10]. Some of the nanocarriers’ benefits include increased permeability through cell membranes and improved protection of the drugs against physical and chemical degradation. Furthermore, nanocarriers improve the therapeutic potential by optimizing drug properties such as stability, solubility, and bioavailability [1,11].

A common strategy for cancer therapy based on the association of multiple chemotherapeutic agents has been implemented as the standard first-line treatment of various malignancies to improve clinical outcome [2]. This approach has shown great potential, particularly to solve the issue of MDR in cancer cells [12,13] and improve anticancer efficacy [14,15,16]. Nonetheless, the administration of multiple drugs is frequently challenging, as different pharmacological agents have distinct pharmacokinetic profiles, resulting in an uncoordinated uptake by the tumor cell, affecting the expected synergistic effect [17,18]. Since nanocarriers can deliver multiple pharmacological agents to the same tumor cell in a single vehicle, the administration of combined drugs utilizing nanocarriers offers the most recent and most efficient therapy for several cancers. The “same time at same place” strategy is appealing since it may increase therapeutic efficacy while minimizing damage to healthy cells through pharmacological synergism, overcoming MDR, and reducing the effective doses [2,19]. Additionally, due to the importance of minimizing harmful side effects to healthy cells, the pharmaceutical market has been more receptive to lipid-based nanocarriers as they are classified by the FDA as generally recognized as safe (GRAS). Lipid-based nanocarriers are also regarded as safe because they are biodegradable and will not accumulate in the body [20].

There is currently a growing interest in the use of natural products in cancer prevention and therapy. Natural compounds and their derivatives have been clinically researched for their capacity to reverse, inhibit, and prevent cancer progression [21]. Due to their proven efficacy in a wide range of malignant tumors with minimal side effects and toxicity, some authors demonstrated that these agents may be a promising option for combination therapy [22].

For their prospective therapeutic applications, nucleic acids such as plasmid DNA (pDNA), small interfering RNA (siRNA), and micro-RNA (miRNA) have been developed into potent tools. Since nucleic acids are able, among other effects, to modulate the expression of genes responsible for MDR, associating chemotherapeutics with nucleic acids has been suggested as an appropriate strategy to increase the effect of cancer therapy [23,24,25,26]. The combination of natural compounds and nucleic acids is a less well-known strategy that has the potential to be very effective as a therapeutic modality that acts by different mechanisms. This combination can lead to a synergistic improvement of the therapeutic effect, a sensitization of the cancer cells to the anticancer activity of the natural compound, and a synergic effect against MDR that restores the anticancer effect.

This review provides a comprehensive overview of lipid-based nanocarriers used for the co-delivery of natural compounds either with chemotherapeutic drugs or with nucleic acids. The utilization of such co-delivery systems offers several benefits, including synergistic/additive/potentiation effects, sensitization of cancer cells, overcoming of MDR, and reduction in adverse effects. Given their promising features, there is an increasing number of reviews exploring the use of natural compounds in cancer treatment (e.g., [27,28,29,30,31,32,33]). However, to date, there has been no comprehensive investigation into the use of lipid-based nanocarriers for the co-delivery of natural compounds and nucleic acids, nor have there been any examples provided of the use of lyotropic liquid crystalline nanoassemblies (LLCNs). Recently, advanced lipid mesophase delivery systems have emerged as a promising class of nanocarrier system. These systems have the potential to encapsulate various cargos with a wide range of lipophilicity properties, making them one of the most advantageous co-delivery systems for cancer [34]. For these reasons, a thorough and up-to-date overview of the studies currently available in the literature is still lacking. Lastly, the review examines the potential opportunities and challenges associated with the implementation of nanocarriers for co-delivery of natural compounds and/or chemotherapeutic drugs and nucleic acids in a clinical context.

## 2. Natural Compounds: Advantages of Combination Therapy in Cancer

Conventional therapy has evident benefits in cancer treatment; however, despite the continuous emergence of new anticancer agents, the majority of chemotherapy-based treatment continues to remain ineffective due to an array of factors, which include chemotherapy-induced toxicity and adverse reactions, insufficient target specificity, and, most importantly, drug resistance during cancer progression (Figure 1) [9].

In this regard, combination therapy has recently become an emerging strategy for tackling the drawbacks of chemotherapy. Simultaneous delivery of two or more therapeutic agents (chemotherapeutic drugs/natural compounds/nucleic acids) can modify different signaling pathways in cancer cells, providing a synergistic response, improving targeting selectivity, optimizing therapeutic effect, and overcoming MDR (Figure 2) [2,17,35]. Thus, taking benefit of the minimal side effects promoted by natural compounds, there is a tendency to follow the potential strategy of combination therapy [9].

### 2.1. Overcoming Multidrug Resistance

MDR is a mechanism that emerges after cells’ exposure to chemotherapeutic agents and refers to the capacity of cancer cells to become resistant to the agents’ effect and can result in the development of malignant cell metastases [40,41]. The cellular mechanisms of MDR can be divided into two general classes: (i) those that block the delivery of chemotherapeutic agents to their target sites, and include the abnormal vasculature which results in low oral chemotherapeutic absorption, early renal clearance, poor bioavailability, and lower tumor site accumulation; or (ii) those that emerge in cancer cells primarily as a result of genetic and epigenetic alterations and directly affect the efficacy of chemotherapeutic agents, and include apoptosis deregulation, increased repair of drug-induced DNA damage, and, enhanced efflux of chemotherapeutic agents [40,41].

Although a wide range of different factors can contribute to MDR, drug efflux changes are considered the major cause of classical MDR [42]. Drug efflux is enhanced by the overexpression of human ATP-binding cassette (ABC) membrane transporters. These transporters are accountable for removing chemotherapeutic agents from cancer cells. Among the ABC transporters, the multidrug resistance protein (MRP) P-glycoprotein (P-gp) is an ATP-dependent drug efflux pump also referred to as multidrug resistance protein 1 (MRP1) (Figure 1). P-gp, the best-studied drug efflux pump, is a significant contributor to chemotherapy failure [42,43]. Furthermore, it has been reported that resistant cells have significantly greater levels of P-gp, and their overexpression is linked to a poor prognosis in a variety of cancers [44].

P-gp-mediated MDR affects several classes of chemotherapeutic agents, such as anthracyclines (e.g., daunorubicin and doxorubicin (DOX)), taxanes (e.g., paclitaxel (PTX) and docetaxel (DTX)), epipodophyllotoxins (e.g., etoposide), and camptothecins (e.g., topotecan and methotrexate (MTX)). As a result, strategies to reverse P-gp-mediated MDR have been extensively researched since the early 1980s, and three generations of P-gp inhibitors are currently classified [40,41,45]. Despite promising in vitro results, there is not, unfortunately, an irrefutable proof of efficacy for the currently available inhibitors, since various clinical trials have been performed to evaluate their anticancer effect, but no significant improvements have been found [21,40]. The development of an ideal inhibitor is commonly associated with the difficulty of finding compounds with high potency and specificity, and with low intrinsic toxicity. Furthermore, it is difficult to achieve specificity of the inhibitors to the ABC transporters, as well as interactions between chemotherapeutic agents and inhibitors [21].

Consequently, in order to overcome such limitations, researchers have shifted their attention to novel approaches for MDR prevention in cancer. In this regard, natural compounds have emerged as an appealing solution, primarily due to their chemosensitizing capacity [46]. Chemosensitizers are small molecules that can increase the sensitivity of cancer cells to chemotherapeutic agents, and those that act as ABC membrane transporter inhibitors are particularly effective. The main example is inhibitors obtained from natural sources, also known as fourth-generation inhibitors, which can interact with ATP binding sites or act directly at MRP binding sites. Natural inhibitors have the potential to be considerably more successful since they offer the most diverse and innovative chemical scaffolds [21]. Moreover, natural compounds with anticancer properties are widely available, as evidenced by the Naturally Occurring Plant-based Anti-Cancer Compound-Activity-Target Database (NPACT) [47]. The main natural compounds evaluated as chemosensitizing agents are highlighted in Figure 3.

Although a wide range of natural compounds, such as terpenoids, alkaloids, steroids, and saponins (Figure 3), have recently been employed to overcome MDR [46,47], phenolic derivatives and flavonoids have been the most cited and studied. According to in vitro biochemical and pharmacological studies, the majority of flavonoids could modulate ABC transporters by competitively binding to the substrate-binding sites and, as a result, delaying cellular efflux [21]. From these chemical families of natural compounds, resveratrol (RSV), curcumin (CUR), and epigallocatechin-3-gallate (EGCG) are the most promising as they can also directly interact with MDR genes [47]. For example, CUR, a polyphenol found in plants of the genus Curcuma, modulates cancer signaling pathways, primarily by inhibiting the nuclear factor kappa B (NF-kB) pathway, as shown in Figure 4. In more detail, CUR modifies signaling pathways of the apoptosis process, by interfering with X-linked inhibitor of apoptosis protein (XIAP), cell proliferation (cyclin D1, ciclo-oxigenase-2 (COX-2), C-myc), cellular inhibitor of apoptosis protein-1 (CIAP-1), cell metastasis (C-X-C chemokine receptor type 4 (CXCR4), ICAM-I), cell invasion (matrix metallopeptidase 9 (MMP-9)), and angiogenesis (vascular endothelial growth factor (VEGF)) [47]. This natural compound also displays P-gp inhibitory activity by downregulating the phosphoinositide 3-kinases (PI3K)/protein kinase B (Akt) [36].

### 2.2. Synergistic, Additive, and Potentiation Effects

The combination of therapeutic agents can result in the following complementary effects [35,48]: (i) synergistic, when the final effect is greater than the sum of individual agents’ effects, resulting in cooperative targeting of activity regulation but with each agent targeting different sites; (ii) additive, that promotes greater or equal effect to the sum of individual agents’ effect; however, both agents act on the same target or pathway; and, (iii) potentiation, in which one agent can enhance the effect of the other or minimize its side effects by regulating pharmacokinetics and/or pharmacodynamics. Furthermore, when both agents in a combination therapy act on the same pathway or target, an undesirable antagonist effect may occur (i.e., when the resultant therapeutic effect is less than the sum of effects of each agent delivered).

### 2.3. Reducing the Side Effects

Combination therapy may also avoid the toxic side effects that normally affect healthy cells. This could happen if one of the co-delivered agents is antagonistic to the other in terms of cytotoxicity. For example, antioxidant supplementation during anticancer treatment may decrease adverse reactions, primarily due to the prevention of reactive oxygen species (ROS)-mediated injury, without compromising anticancer activity [47].

### 2.4. Decreasing the Effective Chemotherapy Dose

One significant drawback of chemotherapy is the high dose of cytotoxic drugs required to achieve a therapeutic effect, which causes serious side effects. In this context, combination therapy appears to be a promising alternative, since the combination of a natural compound and a chemotherapeutic drug may promote an increase in the cytotoxic effect (due to previously described synergistic, additive, or potentiation effects), improve chemotherapeutic performance, and reduce the effective dose required to achieve the necessary therapeutic outcomes [47].

## 3. Lipid-Based Nanocarriers for the Co-Delivery of Natural Compounds and Other Therapeutic Agents

Classic single-delivery therapy (i.e., single administration of the therapeutic agents in their free form) can be challenging due to several drawbacks, including the presence of highly organized physical, physiological, and enzymatic barriers, which make targeting cancer cells with minimal side effects particularly difficult (Figure 1). Furthermore, the varying physiochemical and pharmacodynamic properties of different agents can limit their successful co-delivery [49]. Thus, as previously stated, nanocarriers are an advantageous option for overcoming these challenges, since they are designed to reach specific organs and act selectively on the target site [1,7,10]. In this regard, several nanocarriers have been widely explored for the delivery of anticancer drugs [5]. Lipid-based nanocarriers present some attractive features such as: non-toxic degradation products, biodegradable matrix, low toxicity, high capacity to incorporate lipophilic and/or hydrophilic compounds, and ability to achieve controlled release of encapsulated therapeutic agent [50,51].

### 3.1. Co-Delivery of Natural Compounds and Chemotherapeutics

Table 1 provides several examples of lipid-based nanocarriers co-encapsulating a chemotherapeutic agent and a natural compound for cancer treatment.

Co-delivery of natural compounds acting as chemosensitizers and chemotherapeutic agents with different or comparable mechanisms of action has been identified as the most promising strategy for overcoming undesirable toxicity and other side effects while improving therapeutic effect [5]. However, this co-delivery is also being investigated as a strategy to treat drug-resistant cancers, because of its ability to interfere with a number of signaling pathways in cancer cells [77]. In this regard, the co-encapsulation of different chemotherapeutic agents, particularly DOX, PTX, and 5-fluorouracil (5-FU), with diverse natural compounds using different lipid-based nanocarriers has been reported (Table 1).

DOX, a potent anthracycline, exhibits a broad-spectrum of anticancer activity [61,78]. In brief, DOX’s anticancer mechanism involves two primary possible pathways: (i) producing ROS that cause DNA damage [79,80] and (ii) intercalating into DNA strands and inhibiting topoisomerase II [81,82]. Despite its high efficacy in cancer treatment, DOX’s clinical application is hampered by severe side effects, the majority of which are caused by non-selective DOX-induced apoptosis in tissues and organs [83,84,85], as well as the development of MDR during chemotherapy [86,87]. One co-delivery approach focuses on the combination of DOX with natural compounds, such as CUR [9,39,56], palmitoyl ascorbate (PA) [59], and oleanolic acid (OA) [58], using different lipid-based nanocarriers in order to obtain a synergistic effect. For example, Barui et al. [56] and Tefas et al. [9] developed liposomes co-encapsulating CUR and DOX, and demonstrated their synergism in inhibiting the proliferation, invasion, and migration of tumor cells [9,40]. Zhao et al. [39] studied the cell proliferation inhibition effect of lipid nanoparticles co-loaded with DOX and CUR. The results confirmed the synergistic effect on apoptosis, proliferation, and angiogenesis of hepatocellular carcinoma (HCC), by the increase in Caspase-3 and Bax/Bcl-2 ratio and the decrease in C-myc and VEGF. In addition to CUR, the co-encapsulation in liposomes of DOX with PA, a lipophilic derivative of ascorbic acid, caused an anticancer synergistic effect [59]. Furthermore, the addition of PA not only improved DOX’s anticancer effects [80], but it also demonstrated that this natural compound can mitigate the tissue toxicity of DOX resulting from oxidation [59], as previously described by Shimpo et al. [88]. Sarfraz et al. [58] explored the effect of a liposomal formulation that co-encapsulated DOX and OA, a natural pentacyclic triterpenoid, in a HepG2 mouse model of HCC. This combination had an anticancer synergistic effect, as well as an antagonistic oxidative effect at the cardiomyocytes level, which reduced DOX cardiotoxicity [58]. Overcoming MDR with flavonoids, such as quercetin (QUER) and baicalein (BCL), is the most widely discussed strategy for increasing anticancer effect in several drug-resistant cell lines [55,89,90]. As an example of this strategy, Liu et al. [55] developed hyaluronic acid (HA)-decorated nanostructured lipid carriers (NLCs) to co-deliver DOX and BCL, and reported a synergistic cytotoxic effect in DOX-resistant MCF-7 breast cancer cells. DOX-QUER co-loaded in a lipid-based nanocarrier was also developed as a promising approach for active targeting with the goal of increasing cellular uptake and toxicity against cancer cells [43,91]. Zhang et al. [43] confirmed that QUER can avoid the MDR effect and that biotin (BIO) enhances P-gp inhibition synergistically, resulting in improved antitumor activity. Furthermore, to overcome MDR, the use of DOX combined with other natural compounds, such as docosahexaenoic acid (DHA) [57], α-tocopherol succinate (TS) [61], and *Brucea javanica oil* (BJO) [62], has been reported in the literature. Mussi et al. [57] proposed NLCs co-loaded with DOX and DHA that increased cytotoxicity activity and penetration of DOX, inferring a bypassing of P-gp bomb efflux. The potential of solid lipid nanoparticles (SLNs) co-loaded with DOX and TS—a vitamin E analogue—to overcome MDR and to increase DOX cytotoxicity have been confirmed by two independent studies [61,92]. Li et al. [62] developed lyotropic liquid crystalline nanoassemblies (LLCNs) co-loaded with DOX and BJO in human breast carcinoma cell lines (MCF-7) that have shown an improved anti-tumor effect [62].

PTX is an antimicrotubule chemotherapeutic agent widely used in cancer treatment [70,74]. In cancer cells, PTX induces apoptosis and, as a mitotic inhibitor of cell replication, interferes with microtubule breakdown, which leads to cell cycle arrest [72]. Nevertheless, due to the development of MDR, the potential application of PTX in several cancers is severely limited [36,93]. PTX is a substrate for MDR1 (i.e., for the P-gp channel) [94], and thus the primary strategy for reversing MDR using combined therapy is the co-delivery of PTX with potential P-gp modulators. Several studies were developed in order to reverse MDR, using different natural compounds, such as borneol (BOR) [54], CUR [55,56], cyclosporine A (CycA) [57] and parthenolide (PTN) [74]. Abouzeid et al. [55] and Ganta et al. [56] demonstrated a synergistic effect in MDR cells of CUR and PTX co-loaded in PEG-PE/vitamin E micelles [55] and nanoemulsions [56]. Indeed, CUR enhanced PTX cytotoxicity by down-regulation of the NF-kB and Akt pathways [55,56] (Figure 4). Tang et al. [69] studied lipid–albumin nanoassemblies (LANs) co-loaded with BOR and PTX to achieve greater cellular uptake and improved anti-tumor efficacy. BOR/PTX LANs significantly increased cytotoxicity and drug accumulation in cancer cells, corroborating its potential to enhance the efficacy of chemotherapy. Sarisozen et al. [72] developed actively targeted PEG-PE-based micelles co-encapsulating PTX and CycA, a first-generation P-gp inhibitor, to reverse PTX resistance in P-gp-expressing cells. The authors concluded that the formulation showed a significant increase in cytotoxicity, specifically in drug-resistant cells [72]. Gill et al. [74] studied PTX and PTN co-loaded in micelles that significantly improved anticancer activity against PTX-resistant cell lines. Moreover, the anti-proliferative and pro-apoptotic activity of RSV against MDR tumor cells has been reported [95]. Meng et al. [2] demonstrated that co-encapsulating RSV and PTX in PEGylated liposomes had the potential to reverse PTX-resistance of MCF-7/Adr tumors and improve the efficacy of RSV and PTX, implying their promising use in the treatment of drug-resistant malignancies. The BCL oxidative stress-inducing potential was also considered by Meng et al. [68], who proposed co-loading PTX and BCL in nanoemulsions to enhance antitumor effect and suppress MDR in breast cancer. The antioxidant activity of PA has also improved PTX anticancer activity when they were co-encapsulated in liposomes [73].

5-FU, an antimetabolite, is commonly used to treat colorectal, breast, head, and neck cancers. The anticancer effect of 5-FU is due to the inhibition of thymidylate synthase and the incorporation of its metabolites into DNA and RNA, thereby inhibiting their production [96]. Previous research has shown that RSV synergistically promotes 5-FU-mediated cancer cell apoptosis [52,53]. Mohan et al. [53] investigated the influence of RSV and 5-FU co-loaded in PEGylated liposomes on a head and neck squamous cancer cell line, reporting differential combination effects on gene expression that resulted in cancer cell apoptosis. Furthermore, Cosco et al. [52] evaluated the efficacy of ultradeformable liposomes co-loaded with both agents against squamous cell carcinoma-related lesions. The authors reported that liposomes improved RSV and 5-FU permeation into deeper skin strata, where antioxidant and antiproliferative effects of RSV are essential [52].

### 3.2. Co-Delivery of Natural Compounds and Nucleic Acids

Currently, the efforts to overcome the drawbacks of traditional cancer treatment are mostly focused on strategies that can block the efflux pump effects generated by long-term pharmacological therapy [97]. The combination of natural compounds with nucleic acids is another promising option for a co-delivery system [17]. Furthermore, this is a desirable method for cancer treatment able to overcome MDR and generate synergistic apoptotic effects while reducing toxicity and other side effects [98]. Given their multifunctionality and ability to encapsulate drugs, nanocarriers are the most widely used drug delivery systems [99]. However, many of these delivery systems suffer from non-degradability, complexity, and insufficient biological activity [100]. Lipid-based nanocarriers developed for the co-delivery of nucleic acids and natural compounds (Table 2) are promising due to their low toxicity, biocompatibility, and ease of scaling up [101].

Co-delivery of natural compounds and pDNA [103], siRNA [102,104], or miRNA [38], has been reported and the expression of specific genes can be restored, upregulated, downregulated, or even silenced depending on the type of nucleic acid used [109].

pDNA are small DNA molecules that can carry a gene that will be transcribed into a specific protein of interest, thus improve or restore its function and consequently, different cellular pathways [110]. For example, Xu et al. [103] studied the potential of lipoplexes (i.e., nucleic acids condensed by liposomes) for the co-delivery of p53 pDNA and RSV. The developed formulations were able to up-regulate p53, and the combination of two therapeutic agents demonstrated an anticancer synergistic effect by cell growth inhibition [103].

In comparison to pDNA, miRNA and siRNA act via RNA interference mechanisms.

miRNA can direct and regulate the expression of multiple genes encoding proteins involved in different cellular pathways, both at the transcriptional and translational levels [111]. Thus, Abtahi et al. [38] studied nioplexes (i.e., nucleic acids condensed by niosomes) for co-delivery of CUR and miR-34a, one of the p53 network members. The results showed that combining miR-34a and CUR enabled a synergistic effect, allowing for a reduction in NF-kB expression (Figure 4) and a consequent increase in p53 expression [38].

siRNA has been widely used to selectively down-regulate abnormal protein expression in tumor cells, which is a promising strategy for preventing disease progression [102,106]. The combination of siRNA with different natural compounds, such as gambogenic acid (GNA) [102], CUR [38,104,105,106,107], and gedunin (GED) [108], using lipid-based nanosystems has also been reported. GNA, isolated from Gamboge, is considered a potential anticancer compound as it regulates the expression of cyclin D1 and COX-2 [112,113,114]. Yu et al. [102] studied lipoplexes for co-delivery of VEGF-siRNA and GNA to improve anticancer efficiency in HepG2 cells. According to this study, VEGF-siRNA seemed to mediate VEGF silencing, and the combination with GNA enhanced cell sensitivity and promoted apoptosis [102]. GED, a tetranortriterpenoid isolated from the Indian neem tree, is a Hsp90 inhibitor that demonstrated anti-proliferative effects in several cancers [115,116]. Rana et al. [108] developed lipoplexes for the co-delivery of GED and P-gp siRNA, to enhance the inhibition of breast cancer stem cell proliferation by modulating P-gp and cyclin D1 as well as apoptosis-related genes [108]. CUR has received considerable attention in cancer treatment as the bioactive compound most co-loaded with siRNA. Anup et al. [104,105] developed lipoplexes co-loaded with CUR and STAT3 siRNA. The authors demonstrated that the lipoplexes administrated iontophoretically showed similar efficiency in inhibiting tumor progression and STAT3 protein suppression as intratumorally administration. The authors also reported a synergistic effect of CUR and STAT3 siRNA in cancer cell inhibition [104,105]. Muddineti et al. [106] studied micelleplexes (i.e., nucleic acids condensed by micelles) for co-delivery of CUR and siRNA in a time-dependent manner via a clathrin-dependent endocytosis mechanism [106]. Jia et al. [107] developed CUR and siCCAT1 co-delivered in lipopolyplexes (i.e., nucleic acids condensed by liposomes containing polymers) for colorectal cancer therapy. The results confirmed the ability of the lipopolyplexes to perform endosomal/lysosomal escape efficiently due to the proton sponge effect of the polymer component. Furthermore, the co-delivery of CUR and siCCAT1 could effectively silence CCAT1 and achieve a synergistic effect, thereby enhancing B-cell limphoma-2-mediated apoptosis in human colon cancer cells [107].

### 3.3. Challenges and Opportunities of Co-Delivery Strategies

Although co-delivery of natural compounds and other therapeutic agents in a single nanocarrier is a promising strategy for cancer treatment, its clinical achievement is restricted due to challenges occurring at different stages of the nanocarrier’s development including loading capacity, stability, pharmacokinetics, tumor targeting efficiency, pharmacodynamics, and toxicity [32,117,118,119,120,121]. The first challenge is to choose an appropriate nanocarrier composition capable of simultaneously loading natural compounds and other therapeutics with different physicochemical properties and able to establish distinct chemical interactions. In particular, the co-delivered therapeutics’ association with the nanocarrier should be strong enough to ensure their in vivo stability, avoiding interaction with serum proteins and, as a result, an early release and poor tissue distribution. In this regard, the most striking finding is that LLCNs have not been chosen more often for combination therapy in cancer and co-delivery purposes of therapeutic agents and natural compounds. Indeed, LLCNs merit further investigation and represent a future opportunity for combination cancer therapy purposes, as evidenced by previous studies [37,62,122], because their periodic lipid membranes and networks of aqueous channels present in the inner liquid crystalline organization provide benefits for the entrapment, solubilization, and protection of one or more therapeutics. The internal structure of these nanocarriers, which is formed of liquid crystalline lipid bilayers stacked in precise lattice layouts to form complicated three-dimensional networks of aqueous channels, is what makes LLCNs topology unique. Therefore, LLCNs exhibit a substantially greater surface area when compared to other lipid-based nanocarriers and can encapsulate higher quantities of hydrophobic and hydrophilic therapeutics at sizes equivalent to those of other lipid-based nanocarriers.

Another critical challenge is combining distinct pharmacokinetic behaviors of individual therapeutics, which can result in inconsistent in vivo biodistribution after their co-delivery. As a result, a strict modulation of the nanocarrier is required for precise control of the dose and chronological sequence of each co-delivered therapeutic release. Furthermore, one of the primary goals of co-delivery is to increase the therapeutics’ potency through synergistic or additive effects. However, this goal can be jeopardized by suboptimal therapeutic doses at tumor tissues, so active targeting strategies are required but difficult to implement in order to simultaneously fulfill the different targeting sites of the combined therapeutics. At a pharmacodynamic level it is also challenging to define concentration-dependent effects of natural compounds, and their co-delivery with other therapeutic agents may also produce antagonistic effects. Finally, a major challenge to therapeutic efficacy and clinical translation is that co-delivered therapeutics may cause synergistic systemic toxicity and, as a result, unexpected adverse effects when combined. Therefore, hypersensitivity reactions, long-term toxicity evaluation, and biosafety of nanocarriers and their cargo need to be considered, and only after these issues are fully addressed will co-delivery therapy be widely available in the clinic.

Despite all the above challenges, most co-delivery-based nanotherapeutics have been developed empirically by trial-and-error strategies instead of rational development programs and quality-by-design approaches [120,123]. Few research studies have attempted to comprehensively examine co-delivery combinations using proper characterization and analytical techniques throughout the development process. The characterization of co-delivery-based nanotherapeutics is required from a physicochemical point of view to comprehend its corresponding biological behavior (e.g., knowing how nanocarrier size and shape can be tailored for improved hemodynamics or how to modulate the innate immunity to reduce nanocarrier clearance), and to provide guidance for the process control and safety assessment. In terms of the quantity of parameters necessary for an accurate and thorough characterization, this categorization does not meet consensus. The adoption of reference nanomaterials and internationally recognized procedures is suggested as the solution to unifying the many viewpoints on this subject [117,118]. A characterization of co-delivery-based nanotherapeutics should ideally take place at several points, from the design stage to the assessment of its in vitro and in vivo performance. Timing, chemotherapeutic and/or natural compound metabolic processes, and biological interaction patterns explored in preclinical models will all have an impact on the final clinical outcome of combined treatments. Therefore, a robust biocompatibility testing program, which typically consists of in vivo studies reinforced with chosen in vitro assays to assure safety, is required for the pre-clinical evaluation of co-delivery based nanotherapeutics.

Another issue in the translation of co-delivery-based nanotherapeutics to market and clinical practice is controlling the production process by identifying the critical factors and technology required for a reproducible and economically viable scale-up. The impact of siRNA nanotechnology on pandemic prevention has been demonstrated by COVID-19 vaccines [124]. However, the clinical applications of gene therapy for cancer treatment are currently limited due to the growing stage of the technology, potential unknown risks, higher costs, and selectivity in treating certain types of cancer [97]. Therefore, if this technology is not mature enough in a single delivery context, it is even further away from clinical translation when nucleic acid and natural compound co-delivery is considered. Moreover, based on current understanding, there is an academic interest in co-delivery of natural compounds with chemotherapeutics and/or nucleic acids for cancer treatment that does not mirror the lack of ongoing clinical trials pertaining to the subject matter. This may be attributed to the emergence of new challenges as well as the expenses involved in the design and development of such intricate formulations. The bench-to-market translation of co-delivery-based nanotherapeutics is also severely constrained by the regulatory framework surrounding clinical application. Therefore, it is imperative to address the regulatory and scientific gaps to facilitate the advancement of nanomedicine as a driving force for future biomedical innovation.

## 4. Conclusions

Conventional cancer therapies are still unable to achieve the desired outcomes due to current limitations related with inefficiency and selectivity. As a result, the development of novel therapeutic strategies to overcome these limitations has become critical. Combination therapy has been extensively explored in this context, since co-delivery of natural compounds and chemotherapeutic agents or nucleic acids can achieve stronger anticancer effects through synergistic/additive/potentiation mechanisms, or by improving selectivity, and overcoming MDR either by inhibition of ABC membrane transporters or interaction with MDR genes.

However, while this strategy provides new therapeutic results, it also introduces several new challenges, such as the need to clearly identify the mechanism behind the enhanced anticancer activity by comparing the co-delivery effect with co-administration effect. It is also required to better define the concentration-dependent effect of natural compounds, as well as to evaluate the improvement of their pharmacokinetic parameters when delivered by lipid-based nanocarriers. Moreover, despite lipid-based nanocarriers being considered biocompatible, the safety of the loaded cargo has also to be addressed, namely by long-term toxicity assessment and by studying the immunogenicity issues that may arise from the nucleic acid conjugation. Furthermore, as far as we know, no clinical trials with nanocarriers co-delivering natural compounds or other therapeutic agents have been conducted, which may be due to these challenges as well as the costs of designing and developing such complex formulations.

Despite the critical points that remain unresolved, the co-delivery strategy of natural compounds and chemotherapeutic agents/nucleic acids is undeniably very promising, especially by further exploring versatile nanocarriers such as LLCNs. We strongly believe that this approach, allied with thorough characterization, rational development, and pre-clinical studies, will fulfill the translation of lipid-based nanocarriers into clinical applications.

## Figures and Tables

**Figure 1 pharmaceutics-15-01317-f001:**
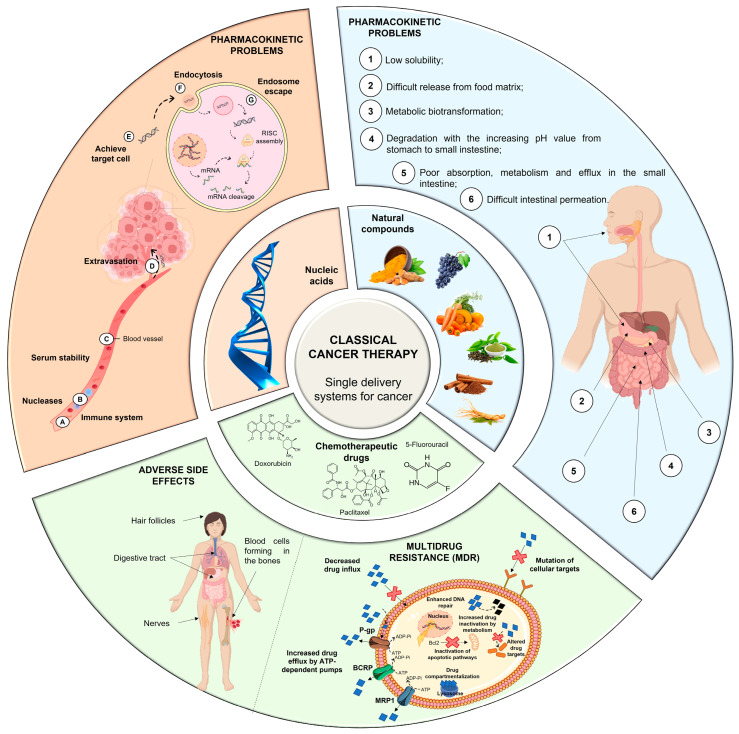
Problems associated with the classical single-delivery therapy (i.e., administration of each of the therapeutic agents in their free form). Nucleic acids, if delivered in the free form, would face different pharmacokinetic challenges, including inactivation by nucleases (A), lack of serum stability due to the immune system (B) and serum proteins (C), extravasation difficulties (D), non-specific distribution in target cells (E), difficulties entering the cell (F), and degradation if not able to escape endosomes (G). Chemotherapeutic drugs, when delivered in the free form, have a nonspecific distribution in cancer cells and healthy cells causing serious adverse side effects, commonly affecting hair follicles, the digestive tract, blood cells and nerves. Furthermore, several MDR mechanisms, such as drug efflux by multidrug resistance protein 1 (MRP1), P-glycoprotein (P-gp), and breast cancer resistance protein (BCRP), or inactivation of apoptotic pathways by B cell leukemia protein (Bcl2), can impair their efficiency. Natural compounds, when administered in their free form, exhibit a number of pharmacokinetic issues that affect their biodistribution and efficacy (1–6).

**Figure 2 pharmaceutics-15-01317-f002:**
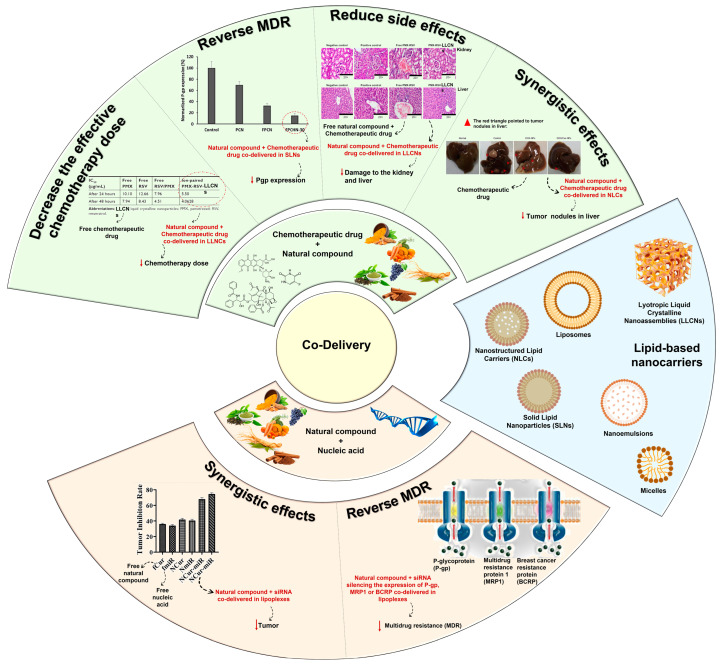
Schematic illustration of lipid-based nanocarriers and advantages of their use for the co-delivery of natural compounds and chemotherapeutic drugs or nucleic acids. Adapted from [36,37], and from [38,39] with permission from Elsevier.

**Figure 3 pharmaceutics-15-01317-f003:**
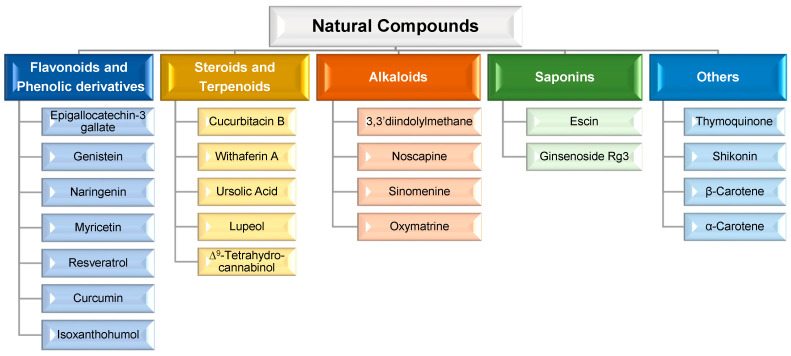
Main natural compounds considered chemosensitizing agents, according to their chemical family [47].

**Figure 4 pharmaceutics-15-01317-f004:**
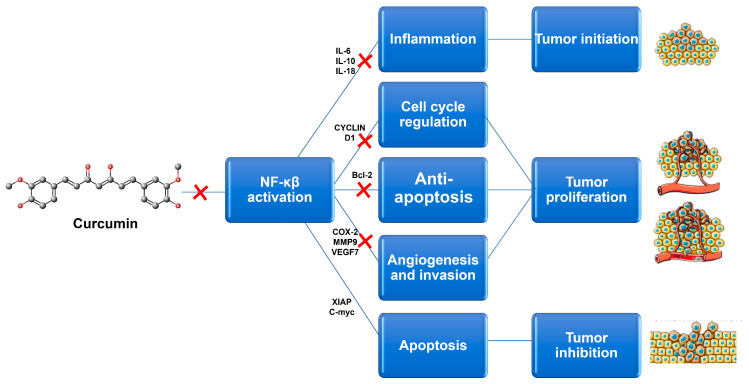
Potential targets associated with Curcumin anticancer activity. This natural compound induces a reduction in its target genes by inhibiting NF-kβ signaling. Bcl-2-B-cell limphoma-2; COX-2-ciclo-oxigenase-2; IL-6-interleukin 6; IL-10-interleukin 10; IL-18-interleukin 18; MMP9-matrix metallopeptidase 9; NF-kβ-nuclear factor kappa B; VEGF-vascular endothelial growth factor; XIAP-X-linked inhibitor of apoptosis protein.

**Table 1 pharmaceutics-15-01317-t001:** Examples of lipid-based nanocarriers co-encapsulating a chemotherapeutic agent and a natural compound for cancer treatment.

Chemotherapeutic Agent	Natural Compound	Lipid-Based Nanocarrier	Composition	Strategy	Ref.
5-FU	RSV	Ultradeformable liposomes	PL90G:SC	Synergistic effect	[52]
		PEGylated liposomes	EPC:DSPE-PEG2000	Synergistic effect	[53]
DTX	CUR	SLNs	Compritol^®^ 888 ATO, GMS, Poloxamer 188Functionalization: Folic acid	Synergistic effect	[54]
DOX	BCL	NLCs	SA, SPC, Precirol^®^ ATO5, Cremophor^®^ ELP, DDAB	Synergistic effect	[55]
	CUR	Liposomes	PEG-RGDK-lipopeptide	Synergistic effect	[56]
		NLCs	Precirol^®^ ATO 5, Labrafac^TM^ lipophile WL 1349, Lipoid S75, Cremophor^®^ RH 40, Glycerin	Synergistic effect	[39]
		Liposomes	DPPC:DSPE:CHOL:PEG2000	Synergistic effect	[9]
	DHA	NLCs	Tween^®^ 80, Oleic acid, Triethanolamine, Compritol^®^ 888 ATO, EDTA	Overcome MDR	[57]
	OA	Liposomes	HSPC:CHOL:DSPE:PEG2000	Synergistic effect	[58]
	PA	Liposomes	PC:CHOL	Synergistic effect	[59]
	QUER	Liposomes	BIO:DSPE:PEG2000	Overcome MDR	[43]
		Phytosomes	Lecithin	Synergistic effect	[60]
	TS	SLNs	Compritol^®^ 888 ATO, TPGS, Triethanolamine	Synergistic effect	[61]
	BJO	LLCNs	GMO:P407 Hexagonal phase inducer: Oleic acid	Overcome MDR	[62]
ETP	CUR	NLCs	GMS, SPC, Oleic acid, DDAB	Decreasing the effective chemotherapy dose	[63]
		Nanoemulsion	SPC, Tween^®^ 80	Additive effect	[64]
GEM	BCL	NLCs	SPC, Precirol^®^ ATO-5, Olive oil, Tween^®^ 80, DDAB	Synergistic effect	[65]
ITC	CTL	Liposomes	DPPC:CHOL:DSPE-PEG2000-FA	Synergistic effect	[66]
MTX	BCN	Lipid–polymer hybrid nanoparticle	DSPE-PEG2000:SA:Gelucire^®^ 50/13:PLA	Synergistic effect	[67]
PTX	BCL	Nanoemulsion	MCT, Soybean oil, Soybean lecithin, Kolliphor^®^ P188, Glycerol	Overcome MDR	[68]
	BOR	Lipid–albumin nanoassemblies	Egg yolk lecithin PL 100 M:BSA	Potentiation effect	[69]
	CUR	SLNs	GMS; TPGS, Tween^®^ 80 Functionalization: Conjugated stearic acid and folate	Overcome MDR	[36]
		Micelles	PEG2000-DSPE/Vit E	Synergistic effect	[70]
		Nanoemulsion	Flaxseed oil, Egg lecithin	Overcome MDR	[71]
	CycA	Micelles	PEG2000-PE	Overcome MDR	[72]
	PA	Liposomes	EPC:CHOL	Potentiation effect	[73]
	PTN	Micelles	PEG2000-DSPE/Vit E	Synergistic effect	[74]
	RSV	Liposomes	PC:DSPE-PEG2000	Synergistic effect	[2]
RAP	BER	Layer-by-layer lipid nanoparticles	GMS, Tween^®^ 80	Synergistic effect	[75]
VNB	Phosphatidylserine	Liposomes	SM:CHOL:DPPS:PEG2000-DSPE	Synergistic effect	[76]
PMX	RSV	LLCNs	GMO:P407 Ion-pairing: CTAB	Reducing side effects	[37]

Table abbreviations: 5-FU-5-Fluorouracil; BCL-Baicalein; BCN-β-carotene; BER-Berberine; BIO-Biotin; BJO-*Brucea javanica oil*; BOR-Borneol; BSA-Bovine serum albumin; CHOL-Cholesterol; CTAB-Cetyltrimethylammonium bromide; CTL-Celastrol; CUR-Curcumin; CycA-Cyclosporine A; DDAB-Dimethyl dioctadecyl ammonium bromide; DHA-Docosahexaenoic acid; DOX-Doxorubicin; DPPC-Dipalmitoyl phosphatidylcholine; DPPS-Dipalmitoyl phosphatidylserine; DSPE-Distearoylphosphatidylethanolamine; DTX-Docetaxel; EDTA-Ethylenediaminetetraacetic acid; EPC-Egg phosphatidylcholine; ETP-Etoposide; FA-Folic acid; GEM-Gemcitabine; GMO-Glyceryl monooleate (or monoolein); GMS-Glyceryl monostearate; HSPC-Dehydrogenated soya phosphatidylcholine; ITC-Irinotecan; LLCNs-Lyotropic liquid crystalline nanoassemblies; MCT-Medium chain triglycerides; MDR-Multidrug resistance; MTX-Methotrexate; NLC-Nanostructured lipid carrier; OA-Oleanolic acid; P407-Poloxamer 407 (or Pluronic^®^ F-127 or Lutrol^®^ F127); PA-Palmitoyl ascorbate; PC-Phosphatidylcholine; PE-Phosphatidylethanolamine; PEG-Polyethylene glycol; PL90G-Phospholipon^®^ 90G; PLA-Polylactic acid; PMX–Pemetrexed; PTN-Parthenolide; PTX-Paclitaxel; QUER-Quercetin; RAP-Rapamycin; RSV-Resveratrol; SA-Stearic acid; SC-Sodium cholate; SLN-Solid lipid nanoparticles; SM-Sphingomyelin; SPC-Soybean phosphatidylcholine; TPGS-α-Tocopherol polyethylene glycol-1000 succinate; TS-α-Tocopherol succinate; Vit E-Vitamin E; VNB-Vinorelbine.

**Table 2 pharmaceutics-15-01317-t002:** Examples of lipid-based nanocarriers co-encapsulating a nucleic acid and a natural compound for cancer treatment.

Nanocarrier	Composition	Nucleic Acid	Natural Compound	Physical-Chemical Characterization			Cancer Cell Lines	Remarks	Ref.
				Size (nm) and PDI	ζ-Potential (mV)	EE (%) and DL (%)			
Lipoplexes	CHEMS, CHOL, PE, PEI	VEGF siRNA	GNA	Size: 200 PDI: <0.3	−30	EE: 81.8 ± 2.04%	HepG2	Downregulation of VEGF expression. GNA loaded lipoplexes have stronger pro-apoptotic effects.	[102]
Lipoplexes	CD014, DOPE	p53 pDNA	RSV	Size: 65.9 to 220.7	+81.4 to +109.8	EE: >90%	MCF-7 and HeLa	RSV and p53 pDNA showed synergistic effect on cells growth inhibition.	[103]
Lipoplexes	DOTAP, DOPE, Sodium cholate, C6 ceramide	STAT3 siRNA	CUR	Size: 157.0 ± 11.0 PDI: 0.46 ± 0.003	+70.5 ± 7.0	EE: 87.5 ± 4.0% (10:1 lipid:CUR ratio)	A431	Downregulation of STAT3 expression. CUR and STAT3 siRNA demonstrated synergistic effect in cancer cell inhibition.	[104]
Lipoplexes	DOTAP, DOPE, Sodium cholate, C6 ceramide	STAT3 siRNA	CUR	Size: 192.6 ± 9.0 PDI: 0.326 ± 0.004	+56.4 ± 8.0	EE: 86.8 ± 6.0%	B16F10	CUR and STAT3 siRNA had a synergistic effect on cancer cell inhibition. The lipoplexes enabled a non-invasive topical iontophoretic application.	[105]
Micelleplexes	Chitosan, Cholesterol chloroformate	siRNA	CUR	Size: 165 ± 2.6 PDI: 0.16 ± 0.02	+24.8 ± 2.2	-	A549	CUR and siRNA were delivered in a time-dependent manner via clathrin-dependent endocytosis mechanism.	[106]
Nioplexes	CHOL, Tween 80, Tween 60, DOTAP	miR-34a	CUR	Size: 60 ± 0.05 PDI: 0.15 ± 0.74	+27 ± 0.30	EE: 100%	A270cp-1, PC3, MCF-7	Co-delivery induced higher cytotoxicity than co-administration.	[38]
Lipopolyplexes	DSPE-mPEG, PEI-PDLLA	CCAT1 siRNA	CUR	Size: 151	+12.37 to −10.48 (depending on CNP:siCCAT1 ratios)	EE: 85.85 ± 3.37% DL: 14.36 ± 1.28%	HT-29	Co-delivery of CUR and siCCAT1 increases HT-29 cell sensitivity to anticancer efficiency of CUR and the silencing effect of CCAT1.	[107]
Lipoplexes	Stearylamine, CHOL, Phosphatidylcholine	P-gp siRNA	GED	Size: 236.01 ± 44.80 PDI: 0.35 ± 0.15	+41.30 ± 4.48	-	MDA-MB 231	Lipoplexes were able to inhibit cell proliferation. Downregulation of P-gp, cyclin D1, p53, Bax, and survivin expression.	[108]

Table abbreviations: A270cp-1-Human ovarian cancer cells; A431-Human skin carcinoma cells; A549-Human lung adenocarcinoma cells; B16F10-Murine melanoma cells; CCAT1-Colon cancer-associated transcript-1; CD014-Peptide-cationic lipid; CHEMS-Cholesteryl hemisuccinate; CHOL-Cholesterol; CUR-Curcumin; D.L.-Drug loading; DOPE-Dioleoyl phosphatidylethanolamine; DOTAP-1,2-dioleoyl-3-trimethylammonium propane; DSPE-mPEG-1,2-distearoyl-snglycero-3-phosphoethanolamine-N-[methoxy(polyethylene glycol); E.E.-Entrapment efficiency; GNA-Gambogenic acid; GED–Gedunin; HeLA-Cervical carcinoma cells; HepG2-Human hepatoma cells; HT-29-Human colon carcinoma cells; MCF-7-Human breast adenocarcinoma (ER-positive) cells; MDA-MB 231-Human breast adenocarcinoma cells with low monocarboxylic acid transporter expression; miR-34a-microRNA-34a; PC3-Human prostate cancer cells; PDLLA-poly (d, l-lactide); pDNA-Plasmid DNA; PE-Phosphatidyl ethanolamine; PEI-Polyethyleneimine; P-gp-P-glycoprotein; RSV-Resveratrol; siRNA-Small interfering RNA; STAT3-Signal transducer and activator of transcription 3; VEGF-Vascular endothelial growth factor.

## Data Availability

Data sharing is not applicable to this article as no new data were created or analyzed in this study.

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
