# Peer review of "Natural Compounds: Co-Delivery Strategies with Chemotherapeutic Agents or Nucleic Acids Using Lipid-Based Nanocarriers"

_pharmaceutics, 2023, doi:10.3390/pharmaceutics15041317_

Round 1
Reviewer 1 Report
The authors present a review on the co-delivery of natural products with the clinicle anticancer chemotherapies and nucleic acids using lipid-based nanocarriers. The literature collection and summary in tables provide useful information to researchers in this field.
Generally, I think the authors can give more details on the challenges of co-delivery. Why this strategy have not been used clinically? It is suggested that the authors including their own experience in the review.
Other issue, the literature style should be checked carefully. For example, some titles are all words capitalized of the first alphabet, some only the first word.
Reviewer 2 Report
The review is relevant for the readership of this journal. However, there are some concerns to be handled .
abstract
Overall, the abstract is not well written. The abstract should have a brief background, scope and approach, key findings, and conclusions. Your abstract is just background, scope, and approach. So, where are the key findings and conclusion?
1. The introduction section needs further improvement with the justification of the novelty of the work.
2. The discussion needs further improvements as well. Especially compare the results in this study with most recent related literature and do a critical evaluation on the trends/similarities/differences.
3. the authors should take some time to edit the entire text to eliminate typo errors.
Reviewer 3 Report
In this review article, the authors compile recent reports involving the use of natural products as chemotherapeutic agents or nucleic acids using lipid-based nanocarriers. Even extensive research is going on in the treatment of cancer however, there are several challenges that need to be addressed. For this, combination therapy has emerged as an excellent strategy for tackling the drawbacks of chemotherapy.
It was enjoyable to read the current review on the association of natural products and nucleic acids in cancer treatment. Therefore, I believe that the current manuscript will be of interest to Pharmaceutics readers and merits publication in the Pharmaceutics in its current form.
Reviewer 4 Report
TThe manuscript is interestig an it show about the development of several strategies to improve the efficiency of chemotherapeutic agents and overcome the conventional therapies challenges against the cancer. One option was co-delivery of the agents in lipid-based nanocarriers provide some advantages by improving the potential of the therapeutic agents carried
It has a good. structure, the figures and tables show the main information and the references are relation with the topic and they are complete
Round 2
Reviewer 2 Report
Thanks for the revision. This is absolutely a better manuscript than the previous one. I recommend this manuscript to be accepted in its current form.